# Electrical performance of a fully reconfigurable series-parallel photovoltaic module

Andres Calcabrini ⊕[1], Mirco Muttillo ⊕[1], Miro Zeman ⊕[1], Patrizio Manganiello ⊕[1] ✉ & Olindo Isabella ⊕[1]

Reconfigurable photovoltaic modules are a promising approach to improve the energy yield of partially shaded systems. So far, the feasibility of this concept has been evaluated through simulations or simplified experiments. In this work, we analyse the outdoor performance of a full-scale prototype of a series-parallel photovoltaic module with six reconfigurable blocks. Over a 4-month-long period, its performance was compared to a reference photo-voltaic module with static interconnections and six bypass diodes. The results show that under partial shading, the reconfigurable module produced 10.2% more energy than the reference module. In contrast, under uniform illumination the energy yield of the reconfigurable PV module was 1.9% lower due to the additional losses introduced by its switching matrix. Finally, a modification in the reconfiguration algorithm is proposed to reduce the output current−voltage range of the module and simplify the design of module-level power converters while limiting the shading tolerance loss.

The role of urban integrated photovoltaic (PV) systems is becoming increasingly important for the development of net-zero-energy districts and the achievement of the UN Sustainable Development Goals[1]. Nevertheless, most PV modules currently available in the market have been designed for utility/scale power plants in open landscapes. While in utility/scale PV power plants PV modules are most of the time uniformly illuminated, PV modules installed in densely populated areas are more frequently exposed to partial shading, which leads to a large reduction in the system's performance ratio[2]. Therefore, in order to facilitate the integration, maximise the energy yield and minimise the levelised cost of electricity of PV systems in the urban environment, PV modules should be designed keeping the shading performance in mind[3–5].

Conventional wafer-based PV modules consist of long strings of series-connected solar cells, aimed to keep the module output current low and minimise Joule losses in cables and power converters. The mainstream approaches to improve the shading tolerance of such PV modules consist in adding bypass diodes[6,7] and connecting groups of solar cells in parallel[8,9]. Diodes are used to bypass the sections of a

string with shaded solar cells allowing the illuminated sections to continue delivering electricity. Meanwhile, parallel interconnections enable shaded and illuminated cells to produce electricity at the expense of a higher output current[10]. In practice, PV modules with parallel interconnections are manufactured with cut cells to limit the module's output current[11,12]. Some of the most noteable commercially available solutions for shading tolerant module topologies include modules with one bypass diode per cell[13,14], series-parallel shingled modules[15,16], and the shingle matrix concept[17,18]. One main advantage of these approaches with parallel interconnections is that the electrical design is generally compatible with the same power converters used for conventional modules. Conversely, the module manufacturing process is often more complex due to the increased number of cells and electrical interconnections per module.

Alternatively, power electronics can be implemented at submodule-level to boost the shading tolerance of PV modules[19,20] without the need of modifying the PV module manufacturing process. While full power processing submodule integrated converters (sub-MICs) have already been implemented in commercially available

[1]Photovoltaic Materials and Devices Group, Electrical Sustainable Energy Department, Delft University of Technology, Delft, The Netherlands.
✉e-mail: p.manganiello@tudelft.nl

solutions[21–23], differential power processing subMICs are an emerging technology that can enable higher power conversion efficiencies[24,25].

In this work, we focus on reconfigurable PV modules as an approach to exploit the benefits of the high shading tolerance of parallel interconnections and the low Joule losses of series interconnections[26,27]. Typically, a reconfigurable PV module consists of two or more blocks of solar cells that are connected to a switching matrix. The switching matrix can dynamically modify the electrical interconnections between the blocks according to the illumination conditions and maximise the module's output power. In previous work, the authors have presented the design of a fully-reconfigurable series-parallel PV module[28–30]. Simulations results, suggest that under partial shading conditions this reconfigurable PV module could boost the annual energy yield by about 10% in comparison to a PV module with fixed interconnections and six bypass diodes. A similar simulation study on an alternative implementation of a reconfigurable PV module indicates that the annual energy yield of reconfigurable PV modules could be between 5% and 15% higher compared to conventional c-Si PV modules with 3 bypass diodes[31].

Although promising, the concept of reconfigurable modules has not yet been validated using actual prototypes in realistic operating conditions. Leveraging previous modelling, design and numerical simulations, we now provide with experimental demonstration that reconfigurable PV modules are capable of outperforming a shade resilient PV module architecture with six bypass diodes. We show the outdoor operation of a highly-performing shade-resilient PV module architecture that does not rely on bypass diodes. Furthermore, we analyse the implications of limiting the degree of reconfigurability of the PV module to simplify the design of a suitable power converter.

This article is organised in the following way. First, we describe the experimental setup used to monitor and compare the performance of the PV modules. Next, we present a direct comparison between the measured performance of the reconfigurable and reference PV modules. Last, we employ submodule current and voltage measurements to analyse the trade-off between shading tolerance and current-voltage output range in the reconfigurable PV module.

## RESULTS
### Experimental setup
The reconfigurable PV module and a reference PV module with static interconnections compared in this study are schematically represented in Fig. 1. Both modules were laminated in-house using commercial 5-inch mono c-Si solar cells from the same batch. Each PV module is divided into six blocks of cells. Each block consists of 16 solar cells connected in series and a bypass diode. In the reference module, all six blocks are (statically) connected in series as illustrated in Fig. 1a. Instead, in the reconfigurable module[28] depicted in Fig. 1b, the blocks of cells are connected to the switching matrix shown in Fig. 1c. It should be noted that the bypass diodes are not required for the operation of the reconfigurable PV module, but they were included in the prototype as an additional safety measure. The switching matrix, implemented with MOSFETs, allows the module to adopt 27 different series-parallel electrical configurations illustrated in Fig. 1d. The 27 configurations are classified as s1p6, s2p3, s3p2 and s6p1, where the first number indicates the amount of blocks that are connected in series forming a string, and the second number indicates how many strings of series-connected blocks are connected in parallel. While s2p3 and s3p2 both represent sets of multiple configurations with the same electrical topology, s1p6 and s6p1 each represent a single configuration where all the blocks are connected in parallel and in series, respectively.

A synchronous reconfiguration algorithm is implemented in a microcontroller to dynamically select the best configuration of the reconfigurable PV module in response to the illumination conditions. The reconfiguration algorithm measures the short-circuit current of the six blocks of cells and aims to connect as many blocks in series as possible without exceeding predefined thresholds that determine the maximum relative current difference for each set of configurations[28] as explained in the Methods section. For instance, when the PV module is uniformly illuminated, configuration s6p1 is chosen since it delivers the lowest current and minimises Joule losses. On the other hand, when the PV module becomes partially shaded, configurations with parallel interconnected blocks will be chosen to reduce current mismatch losses. It should be noted that, from all the configurations, s1p6 is the one that presents the highest shading tolerance, but at the same time the one that implies the highest currents and thus Joule losses.

Both PV modules were mounted on a rack as shown in Fig. 2a at the PVMD group monitoring station (51.9997° N, 4.3690° E) in Delft, the Netherlands. The thermal and electrical performance of the PV modules have been monitored during 4 months under different shading conditions. The experiments were conducted between May and August 2021 to take advantage of the sunniest period of the year in the Netherlands, which was essential to induce partial shading.

During the four-month-long monitoring campaign four shading experiments were performed as illustrated in Fig. 2b–e. During experiment 0, the modules were unshaded and facing South. During experiments 1 to 3, the PV modules were shaded by objects that were symmetrically fixed on the mounting rack to emulate shading caused by structures usually present on rooftops, such as chimneys and dormers. The goal of these experiments was to evaluate the performance of the module when shading different numbers of groups of cells simultaneously. In Experiment 1, the rack was facing South and the shading bars were distributed to shade 3 blocks at the same time. Different bar lengths were used to progressively shade and unshade the blocks during sunrise and sunset. In Experiment 2, the rack was rotated towards Southeast and the shading bars were placed together close to the top of the rack to shade mostly the top left block at noon. In Experiment 3, the shading objects were widened to shade the two blocks on the top left corner of the PV modules at noon.

### PV module performance
The power generated by the reconfigurable and the reference PV modules was compared using the maximum power point (MPP) extracted from the I-V curves measured with a BK8616 electronic load and a LPVO PV monitoring unit in combination with an LPVO MP1010F-2 maximum power point tracking (MPPT) unit, respectively. As an example, the power delivered during a clear sky day by both PV modules during experiment 1 is presented in Fig. 3a. It can be noticed that, when the PV modules were partially shaded (i.e., in the morning and the afternoon) the reconfigurable PV module generated more power than the reference module. During these intervals, the voltage time-series in Fig. 3b reveals the progressive activation of the bypass diodes in the reference module and how the reconfigurable PV module switched between configurations throughout the day to minimise current mismatch losses.

Fig. 3a also shows that at noon, when the Sun was in the South and the modules were unshaded, the reference PV module outperformed the reconfigurable PV module. From Fig. 3b it is clear that both modules had the same electrical layout at noon since the reconfigurable module operated in the s6p1 (i.e., all-series) configuration. The voltage (and power) loss in the reconfigurable module during this period is a direct consequence of the resistive losses in the MOSFETs and the PCB traces of the switching matrix, which amount to an increase of 7.9% in the equivalent series resistance of the solar cells[28].

The DC energy delivered by the PV modules during each of the experiments is summarised in Table 1. The shading row in Table 1 indicates the fraction of time (excluding dark hours) that the irradiance difference between the least and most illuminated cells in the module was higher than 50%, as a proxy for the amount of time that the modules were partially shaded. Results indicate that the

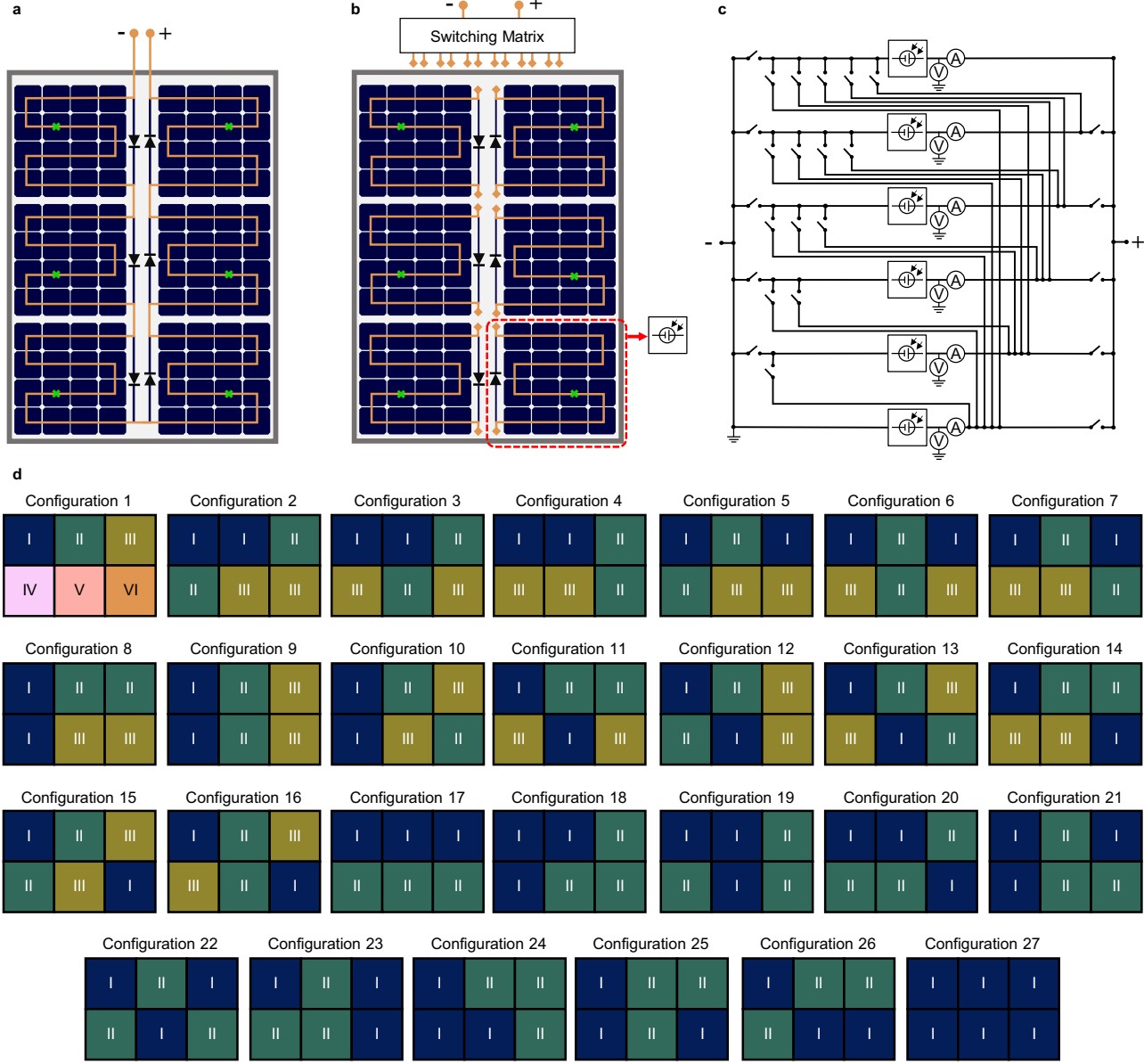

**Fig. 1 | Evaluated PV module topologies. a** Reference PV module (REF) with 96 series-connected solar cells and 6 bypass diodes. **b** Reconfigurable PV module (REC) with 6 blocks, each made of 16 series-connected solar cells. **c** Switching matrix schematic. Switches, current and voltage sensors have been implemented with MOSFETs, Hall sensors and resistive voltage dividers, respectively. Measurements are digitised using a 12-bit ADC and processed by an ARM microcontroller in the switching matrix that executes the reconfiguration algorithm and controls the state of the MOSFETs. **d** All possible electrical configurations of the proposed reconfigurable module. Each coloured square represents a block of cells. Blocks with the same number and colour form series-connected strings which are subsequently connected in parallel. Configuration 1 is the only s1p6 configuration. Configurations 2-16 are the 15 possible s2p3 configurations. Configurations 17-26 are the 10 possible s3p2 configurations. Configuration 27 is the only s6p1 configuration.

reconfigurable PV module generated about 1.9% less energy than the reference module in the absence of shading due to the additional resistive losses in the switching matrix. Instead, under partial shading conditions (i.e., during experiments 1, 2, and 3), the reconfigurable PV module generated between 4.8% and 13.7% more energy than the reference module. It can be noticed that the difference is larger for the experiments when the PV modules were more frequently subject to partial shading (refer to Shading row in Table 1). Moreover, the DC yield difference between the reconfigurable and reference PV modules also depends on the number of blocks that are simultaneously shaded. From the positions of the shading objects in Fig. 2d, it can be noticed that during experiment 2, only one of the top blocks of cells was shaded most of the time. This shading pattern was effectively mitigated by the REF module, where the shaded block was bypassed and only one sixth of the total module power was lost. As a consequence, the DC yield difference in experiment 2 was significantly lower than in experiments 1 and 3, when more than one block of cells was often shaded at the same time.

It is worth mentioning that the yield of the reconfigurable module in Table 1 excludes the energy consumed by the switching matrix and the sensing circuitry, which is considered negligible. While the switching matrix can be kept in low-power consumption mode in between reconfiguration intervals reducing its consumption to 10 mW, the reconfiguration intervals last in average 150 ms[28], during which the power consumption raises to about 1 W. Considering 16 hours of daylight and minutely reconfiguration events, it is estimated that the energy consumed by the electronics in the reconfigurable module was approximately 20 Wh for the entire monitoring campaign, which is

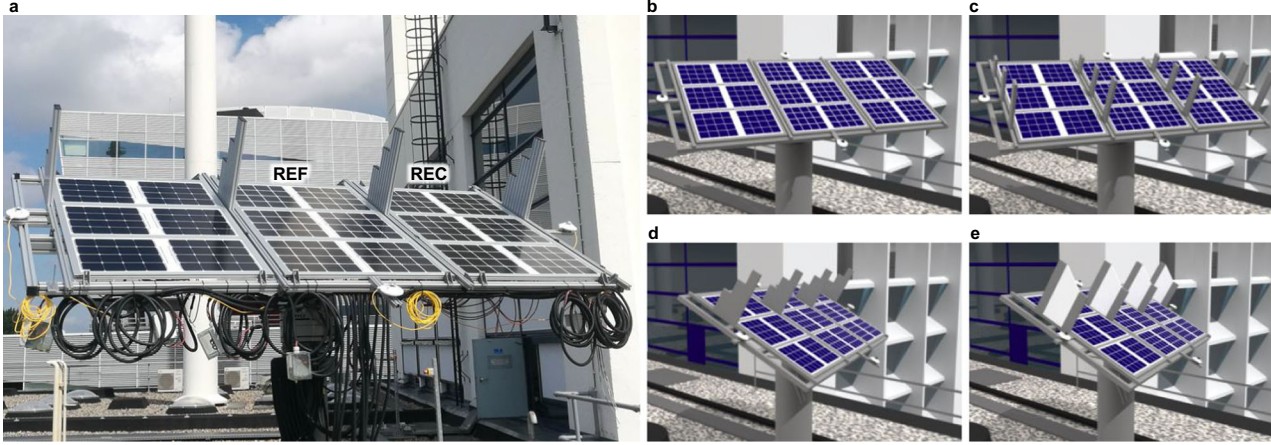

**Fig. 2 | Experimental setup at the PVMD group monitoring station in Delft, the Netherlands. a** Photograph of the experimental setup. The shading objects were placed on both sides of the PV modules to create the same shading conditions. REF and REC stand for the reference module and the reconfigurable module, respectively. **b** 3D model of shading experiment 0. **c** 3D model of shading experiment 1. **d** 3D model of shading experiment 2. **e** 3D model of shading experiment 3. In all four experiments the modules were tilted 30°. During experiments 0 and 1 **b**, **c** the modules were facing South. During experiments 2 and 3 **d**, **e** the modules were facing Southeast.

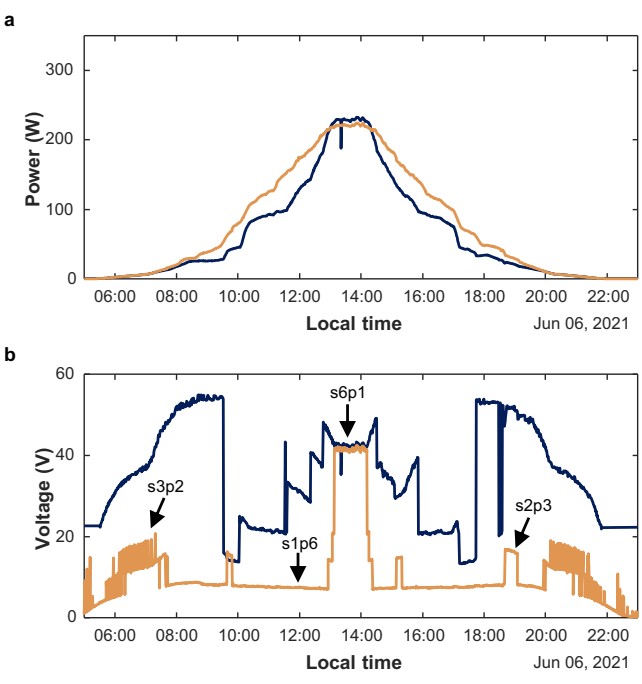

**Fig. 3 | Comparison of the electrical output of the reference (REF) and reconfigurable (REC) PV modules in a clear sky day during experiment 1. a** Maximum power time-series. **b** MPP voltage time-series. The different voltage levels of the REC curve correspond to different types of electrical configurations. Source data are provided as a Source Data file.

about 0.02% of the total energy delivered by the reconfigurable PV module.

The DC yield difference between both modules was also evaluated as a function of the solar position. The results presented in Supplementary Fig. 1 show that in the absence of shading, the reference module performed mostly better than the reconfigurable module during of Experiment 0. Instead, during the experiments when shading objects were mounted on the rack, the reconfigurable module almost always outperformed the reference module except in two particular situations: (1) when the sun was in front of the module

(i.e., around South in Experiment 1, and around Southeast in Experiments 2 and 3), because the shading objects on the sides of the modules were not causing shading, and (2) when the sun was behind the plane of array, because the modules received only diffuse and reflected irradiance and the current mismatch was significantly reduced.

The pie charts in Fig. 4 show the percentage of the energy delivered by the different configurations of the reconfigurable module during each of the shading experiments. As expected, during experiment 0, the reconfigurable PV module was mostly operated in the s6p1 (all-series) configuration because the module was uniformly illuminated. During experiments 1 to 3, the algorithm frequently chose configurations with parallel interconnections to reduce the current mismatch caused by the partial shading. In particular, it should be noticed the s1p6 (all-parallel) configuration contributed to a significant share of the total generated electricity in all three experiments when the PV module was partially shaded. As explained before, the s1p6 configuration implies low output voltages and high output currents that entail a burden on the design and performance of the power converter for the reconfigurable PV module.

## Limiting the electrical operating range

Results show that in the absence of shading, the REC module is mostly operated in the s6p1 configuration and delivers power at approximately the same current as the REF module. Instead, when there is partial shading, the reconfigurable PV module tends to deliver higher currents because the algorithm chooses configurations with different combinations of blocks connected in parallel. The electrical operating ranges of the REF and REC modules are compared in Fig. 5, where it is illustrated how much energy was delivered at each voltage, current and power level during the experiments when the PV modules were partially shaded. In Fig. 5c, it can be noticed that the REC module generated up to 40.8 A, about 6 times more than the maximum current of 7.2 A delivered by the REF module in Fig. 5a. Besides, the output voltage of the REF module ranged from 13.3 V to 57.9 V, while the output voltage of the REC module range from 5.6 V to 52.3 V. As a result of the extended electrical output range, a power converter for a reconfigurable PV module would be larger, less efficient and/or more expensive than a power converter for a conventional PV module. Although the design of a suitable power converter for a reconfigurable PV module is beyond the scope of this work, one way to reduce the burden on the design of such power converter is by adapting the

operation of the reconfiguration algorithm. This specific approach will be addressed in the rest of this Section.

One straightforward approach to ease the design of the power converter consists in limiting the electrical output range of the reconfigurable module. This can be achieved by avoiding the configuration s1p6, which delivers the highest currents and lowest voltages. When configuration s1p6 is avoided, the reconfiguration algorithm chooses as an alternative the best s2p3 configuration, and consequently, the reconfigurable module becomes less shading tolerant.

Even though the best s2p3 configuration was not measured, it can be reconstructed using the I–V curves of the different blocks, which were regularly measured during the whole experimental campaign with the current and voltage sensors integrated in the switching matrix. The I–V curve of the different module configurations were recreated by interpolating and adding the measured voltages (in the case of series connections) or currents (in the case of parallel connections) of each of the six blocks of cells in Fig. 1c. This method for

generating I–V curves has been validated for various PV module topologies[10,28].

Figure 6 depicts the operating range of the reconfigurable PV module for the presented shading scenarios after modifying the

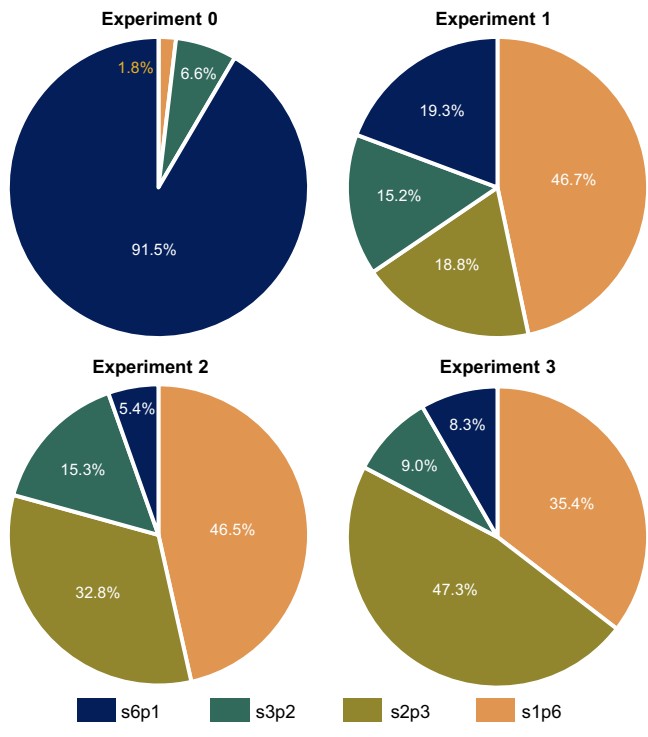

**Fig. 4 | Energy yield share by configuration during each shading experiment.** The start and end dates of the experiments are specified in Table 1. While the s6p1 and the s1p6 represent only one configuration each, the pie sections corresponding to the s3p2 and s2p3 represent multiple possible configurations with similar electrical characteristics as illustrated in Fig. 1d.

**Table 1 | DC energy yield during each experiment**

| Experiment | 0 | 1 | 2 | 3 |
|---|---|---|---|---|
| Start date | June 12th | May 7th | June 26th | August 3rd |
| End date | June 24th | June 10th | July 22nd | August 26th |
| DNI (kWh/m²) | 40.2 | 109.9 | 52.6 | 41.8 |
| DHI (kWh/m²) | 36.0 | 81.3 | 75.7 | 59.6 |
| GHI (kWh/m²) | 76.0 | 190.8 | 127.8 | 101.0 |
| Shading (%) | 0 | 34 | 24 | 30 |
| DC yield REF (kWh) | 16.8 | 32.5 | 22.8 | 18.0 |
| DC yield REC (kWh) | 16.5 | 37.0 | 23.9 | 19.9 |
| DC yield diff. (%) | -1.9% | 13.7% | 4.8% | 10.3% |

Experiments were performed in the year 2021, started at 5 a.m. and concluded at 11 p.m. local time. Shading (%) is calculated as the time that the irradiance on the most shaded cell in the PV module is less than 50% of the irradiance on the most illuminated cell in the PV module divided by the total daylight hours in a year.

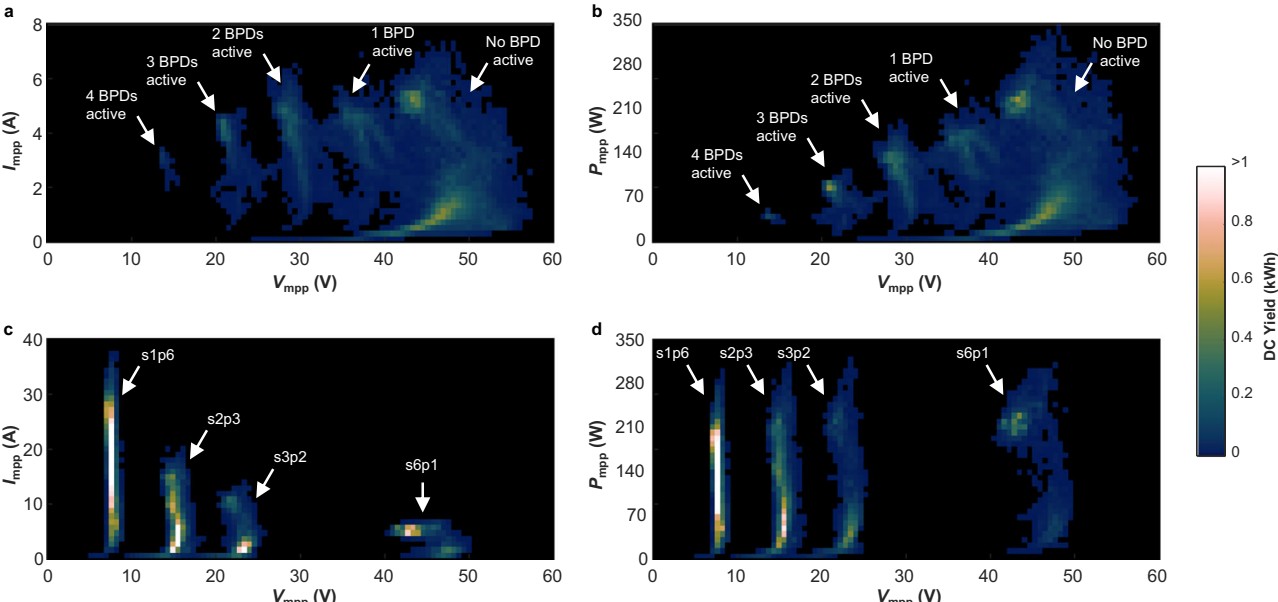

**Fig. 5 | Operating point histogram of the PV modules during the shading experiments 1, 2, and 3. a** Current and voltage delivered by the PV module with 6 bypass diodes (REF). **b** Power and voltage delivered by the PV module with 6 bypass diodes (REF). **c** Current and voltage delivered by the reconfigurable PV module (REC). **d** Power and voltage delivered by the reconfigurable PV module (REC). The start and end dates of each experiment are specified in Table 1. The colour bar indicates how much energy was delivered at the different operating points. Source data are provided as a Source Data file.

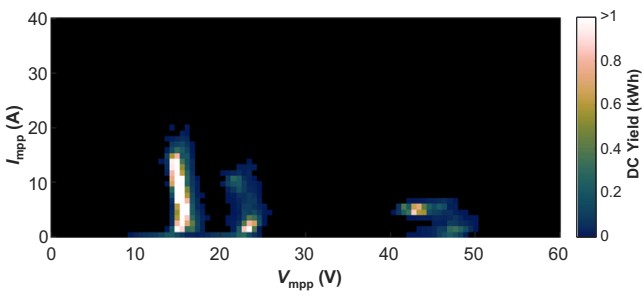

**Fig. 6 | Operating point histogram of the reconfigurable PV module during the shading experiments 1, 2, and 3 after modifying the reconfiguration algorithm to avoid the s1p6 (all-parallel) configuration.** The colour bar indicates how much energy was delivered at the different operating points. Source data are provided as a Source Data file.

algorithm to preclude configuration s1p6. While the REC module with the original reconfiguration algorithm generated in average 10.2% more energy than the REF module (only considering shading experiments 1 to 3), the energy yield of the REC module with the modified algorithm would be 6.4% higher than the REF module. In return, the maximum output current of the reconfigurable PV module would drop to 20.2 A and the output voltage would range from 10.7 V to 52.3 V.

## Discussion

Our measurements indicate that in the absence of shading the reconfigurable PV module performed 1.9% worse than the reference module due to the additional resistive losses introduced by the switching matrix. Instead, when the PV modules were subject to partial shading, the reconfigurable PV module delivered an average 10.2% more energy. In particular, the difference between the energy yield of both modules was larger for shading scenarios in which two or more blocks of cells were often shaded at the same time.

During all the shading experiments, about 40% of the energy was delivered by the s1p6 configuration, in which all six blocks of cells are connected in parallel. Although the s1p6 is the most shading tolerant of the 27 possible configurations, it is also the one that generates the highest currents and thus may lead to the highest losses at system level. To avoid high currents, we analysed the performance of a hypothetical reconfigurable PV module in which the configuration s1p6 is precluded. This analysis was performed by using the I–V curves of each block of cells, that were regularly measured during the mentioned shading experiments, to reconstruct the I–V curve of different module configurations. On the one hand, we show that this approach limits the maximum current and the minimum voltage generated by the reconfigurable PV module, which implies a simpler design of the power converter and reduced Joule losses in cables. On the other hand, it was found that when the s1p6 configuration is precluded, the yield of reconfigurable module would be reduced on average by 3.8%. It is expected that by reducing the output current of the PV module to approximately 20 A, Joule losses in cables and conversion losses in power electronic devices would also be significantly smaller. In order to verify whether it is beneficial to avoid configuration s1p6, the complexity of the design and efficiency of power converters for both versions of the reconfigurable module must be carefully investigated and system-level analyses must be performed.

In the future, asynchronous algorithms together with other sensed parameters at the level of the PV module can be exploited to improve the performance of reconfigurable modules. Most interestingly, an approach based on machine learning could be implemented to facilitate an AI engine which optimally controls the reconfiguration. Finally, the presence of a microcontroller in the envisioned smart junction box can also be used to tokenise and trade energy packets in the future digital energy market.

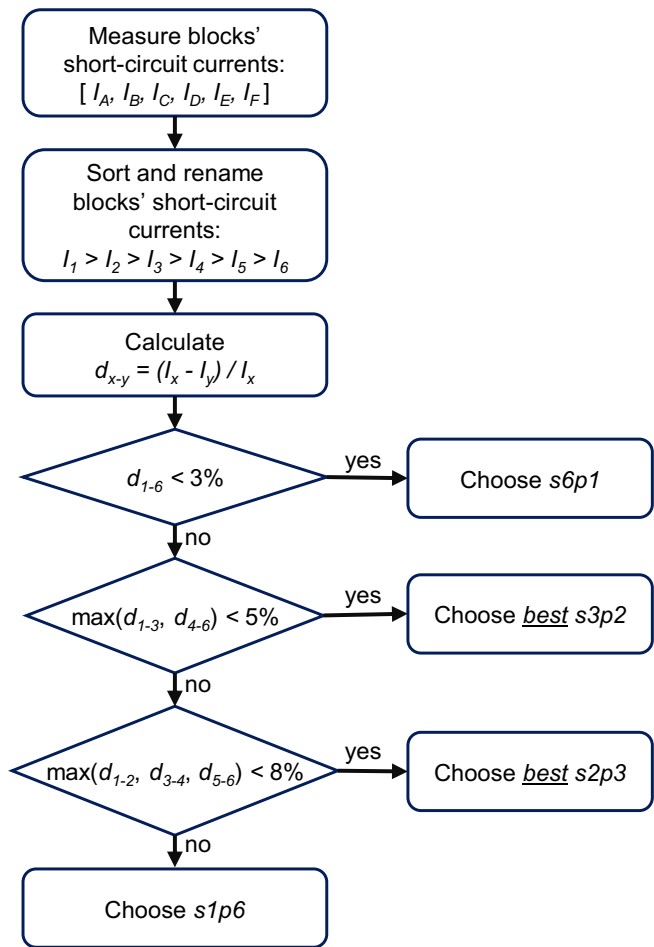

**Fig. 7 | Reconfiguration algorithm.** The algorithm was executed every minute and the threshold values chosen to determine the configurations were optimised to maximise the DC energy yield.

## Methods

Each of the blocks of 16 series-connected solar cells, that form both the reference and reconfigurable PV modules, were laminated in-house using the materials listed in Supplementary Table 1. The electrical characteristics of the solar cells are given in Supplementary Table 2 and, after lamination, the I–V curve of each block was measured using an EternalSunSpire A+A+A+ Xenon Single Long Pulse flash simulator to check the uniformity of the I–V characteristics of the blocks of cells under AM1.5g illumination.

The temperature of the modules was measured using T-type thermocouples attached to the backsheet of the PV modules and thermally insulated from the air at the positions indicated with green crosses in Fig. 1a, b.

The electrical performance of the reference PV module (REF) was monitored using an LPVO PV monitoring unit (PVMU) in combination with an MP1010F-2 MPPT unit[32]. The PVMU was used to perform I–V sweeps on the reference PV module at every minute. Meanwhile, the MP1010F-2 MPPT unit was employed in between these sweeps, ensuring a continuous extraction of power from the reference module by operating it at its maximum power point.

The electronics required to operate and monitor the reconfigurable PV module were installed in the electrical box shown in Supplementary Fig. 2, which was mounted on the rear side of the rack. The microcontroller unit, where the reconfiguration algorithm shown in Fig. 7 was implemented, controls the state of the MOSFETs of the switching matrix through a driver circuit.

In this prototype, all the devices were powered by an external power supply. Nevertheless, the power consumption of the switching matrix is minimal and it can be eventually self-powered by the solar cells in the PV module. The electrical performance of the reconfigurable PV module (REC) was monitored using a BK8616 (B&K PRECISION) electronic DC load which was connected to the output of the switching matrix. Both the REC and the REF PV modules were connected to the respective measurement equipment using 4-wire connections. Although it was not possible to monitor both modules using the same equipment due to the differences in the electrical output ranges, the selected equipment has similar measurement accuracy as shown in Supplementary Table 3. The maximum measurement error was 7.5 mW and 5.5 mW with the BK8616 and the LPVO PVMU, respectively.

In addition, the voltage and current sensors integrated in the switching matrix (see Fig. 1c) were calibrated to measure the voltage and current of each individual block. Only the current sensors were required for the operation of the reconfiguration algorithm. The voltage sensors were added to enable the measurement of block-level $I-V$ curves, which were then processed to emulate different operating conditions.

The measurement procedure for the reconfigurable module is shown in Fig. 8. The setup consisted of a PC that was used to synchronise the operation of the switching matrix and the DC load, and to store the measurements. Reconfiguration was performed on a minutely basis by the microcrontroller in the switching matrix upon the reception of the triggering command sent by the PC. Before each reconfiguration event, all the blocks in the reconfigurable PV module are connected in parallel to measure the $I-V$ curves of each block using the current and voltage sensors integrated in the switching matrix. Next, the microcontroller of the switching matrix executes the reconfiguration algorithm and the reconfigurable module switches to the optimal configuration. Finally, the $I-V$ curve of the reconfigurable module is measured with the electronic DC load and then the module is kept at its maximum power point using a perturb and observe (P&O) maximum power point tracking algorithm, also implemented through

the BK8616 electronic load, for 1 min until the next reconfiguration event. It should be noted that under normal operation, it is not necessary to measure the $I-V$ curves of each block of cells because the reconfiguration algorithm only requires the short-circuit current to find the best module configuration[28].

## Reporting summary
Further information on research design is available in the Nature Portfolio Reporting Summary linked to this article.

## Data availability
The PV module performance data generated in this study have been deposited in the 4TU.ResearchData data repository [https://doi.org/10.4121/52702ca1-87ca-4429-b3ec-11cf2eb1c921][33]. Source data are provided with this paper.

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

## Acknowledgements
This work is supported by the sector plan of the Dutch government in photovoltatronics research. Furthermore, the authors would like to thank EternalSun Spire (www.eternalsunspire.com), especially Stefan Roest and Elias Garcia Goma for helping us to test the PV modules at EternalSun's facilities.

## Author contributions
Conceptualisation: A.C., M.M., P.M and O.I.; Methodology: A.C., M.M. and P.M.; Hardware: M.M. and A.C.; Software: M.M. and A.C.; Investigation: A.C., M.M. and P.M.; Resources: P.M., O.I. AND M.Z.; Writing - Original Draft: A.C.; Writing - Review & Editing: M.M., P.M., O.I. and M.Z.; Supervision: P.M., O.I and M.Z.

## Competing interests
The authors declare no competing interests.
