## [Peer Review File · Nature Communications]

Electrical performance of a fully reconfigurable series-parallel photovoltaic moduleREVIEWER COMMENTS

Reviewer #1 (Remarks to the Author):

In this paper, the authors present results of the experimental test of a series-parallel module with six reconfigurable blocks in partial shading condition. The added value of the proposed technology is clearly demonstrated. Limitations of the approach notably in shadow free conditions and due to the high ohmic losses of the fully parallel configurations are discussed and a solution is proposed. The article is very well written and comprehensive. Possible suggestions for improvements are listed below:

- Just as a remark: in the first paragraph of introduction, the authors implicitly argue that shadow is not an issue for large PV system because they are installed open landscape. This is not always the case due to surrounding orography and forest but, more importantly, large PV plants are very often impacted by self-shadow as reported in <https://doi.org/10.1016/j.solener.2019.03.086>
- The choice of the four experiments could be further motivated: are the shadow profiles and orientations representative for a particular environment?
- In table 1, the length of each test is provided but giving and discussing the period of the year when the experiments have been conducted is also important because it is related to the sun position as well as the cloudiness.
- It would be very interesting to differentiate Fig. 8 for each experiment unless the authors motivate the reasons for gathering the results in a single figure.
- In Figure 8, the representation in a I – V space is very good and should be kept but I would recommend adding the same figure in a plot representing the power as a function of the voltage.
- Finally, an analysis of the gain as a function of the sun position (solar azimuth and elevation angles) would allow the reader getting a better insight about the operation and gain of the reconfigurable series-parallel PV module.

Reviewer #2 (Remarks to the Author):

The proposal is important and very interesting. The paper is well written, but some points should be considered:

- What is the LPVO MP1010F-1? I could not find in the reference [21] a full description of the hardware. Is it a dc-dc converter or what? More technical details are necessary.
- The reference module is connected to the LPVO and the reconfigurable module to a DC load. For sake of comparison is very important to use the same system and same measurements. The behavior of the MPPT, for example, can affect considerably the results.
- Furthermore, in order to use a single module in a real application it is necessary a dc-dc stage in the inverter. This can affected the measurements and the behavior of the reconfigurable matrix. For a definitive validation of the proposal, it is important to use a system similar to the one that will be used in actual applications.
- It is said that a microcontroller is used for the reconfiguration algorithm, but according to Fig. 4 it seems that a PC is used.
- Even if the energy consumed by the microcontroller, sensing circuitry, mosfets gates, etc. is low, it must be considered in order to a fair comparison
- The paper is as an experimental validation of [17]. [17] is much more complete and some important information could be put in the present paper

The work is good, but I think that the validation should be improved for a publication on this high impact journal.

Reviewer #1

The Authors wish to thank the Reviewer for their valuable comments and recommendations aimed at improving the quality of the manuscript.

A detailed reply to the comments of Reviewer #1 follows here below. Reviewer's comments are in black colour whereas Authors' replies are in red colour.

1. In the first paragraph of introduction, the authors implicitly argue that shadow is not an issue for large PV system because they are installed open landscape. This is not always the case due to surrounding orography and forest but, more importantly, large PV plants are very often impacted by self-shadow as reported in <https://doi.org/10.1016/j.solener.2019.03.086>

In the referred paragraph it is mentioned that PV modules in large PV plants are most of the time uniformly illuminated. This statement is true for well designed PV power plant with reasonable ground cover ratio values, even if the installation is on a complex terrain with shading objects like trees, substations, and high voltage electric poles. However, this does not mean that partial shading is not an issue, because in most practical cases row-to-row shading is inevitable in the early morning and late afternoon. However, the approaches for minimizing the impact of partial shading in large PV plants (e.g. special stringing considerations, more MPPT channels per inverter, backtracking) are very different from the methods described in paragraphs 2 and 3, which are applicable to PV systems in urban environments.

To prevent readers from misinterpreting that partial shading is not present in large power plants, the authors have added the word "more" in the first paragraph to emphasize the fact that PV modules in large power plants are also exposed to partial shading.

2. The choice of the four experiments could be further motivated: are the shadow profiles and orientations representative for a particular environment?

The following text has been added at the end of the Experimental setup subsection:

[...] During experiments 1 to 3, the PV modules were shaded by objects that were symmetrically fixed on the mounting rack to emulate shading caused by objects and structures usually present on rooftops, such as chimneys and dormers. The goal of these experiments was to evaluate the performance of the module when shading different numbers of groups of cells at the same time. In Experiment 1, the rack was facing South and the shading bars were distributed to shade 3 blocks at the same time. Different bar lengths were used to progressively shade (and unshade) the blocks during sunrise (and sunset). In Experiment 2, the rack was rotated towards Southeast and the shading bars were placed together close to the top of the rack to shade mostly the top left block at noon. In Experiment 3, the shading objects were widened to shade the two blocks on the top left corner of the PV modules at noon. [...]

3. In table 1, the length of each test is provided but giving and discussing the period of the year when the experiments have been conducted is also important because it is related to the sun position as well as the cloudiness.

The authors agree that sun position and cloudiness are essential to determine when a PV module is subject to partial shading. However, these factors are difficult to quantify and to relate to the performance of the module in different geographical locations. Instead of focusing on discussing these factors, the authors have included DNI, DHI, GHI and the shading percentage for each experiment in Table 1. The shading factor incorporates the effect of the solar position and the ratio between the diffuse and direct irradiance caused by cloud cover in a way that is more relatable to other locations. The following sentences have been added in to clarify this point:

[...] The experiments were conducted between May and August 2021 to take advantage of the sunniest period of the year in the Netherlands, which was essential to induce partial shading. [...]

[...] The “shading” column in Table 1 indicates the fraction of the time (excluding dark hours) that the irradiance difference between least and most illuminated cells in the module was higher than 50% as a proxy for the amount of time that the modules were partially shaded. [...]

The measured irradiance components and ambient temperature during the experiments have been included in the public repository referred to in the Data availability section.

In addition, we have added an analysis as a function of the solar position in response to the reviewer’s last point.

4. It would be very interesting to differentiate Fig. 8 for each experiment unless the authors motivate the reasons for gathering the results in a single figure.

The goal of Figure 8 is to show the operating region on the reference and reconfigurable PV modules independently from the shading conditions. The reason why Figure 8 is not arranged by experiment is because the differentiation by experiment does not provide much additional information to the one already given in the manuscript. The figure below shows that during the 3 experiments the voltage and current range of the reconfigurable module were approximately the same. When differentiating by experiment it is possible to notice that during Experiment 1, the reconfigurable module operated more often in configuration s6p1 compared to Experiments 2 and 3. Also, it can be seen that during Experiments 2 and 3 it operated more often in configuration s2p3 compared to Experiment 1. In the authors’ opinion these two observations are difficult to see in the plots below whereas they are more clearly illustrated in the pie charts of Figure 7.

5. In Figure 8, the representation in a I – V space is very good and should be kept but I would recommend adding the same figure in a plot representing the power as a function of the voltage.

Figure 8 has been modified following the reviewer’s suggestion.

6. Finally, an analysis of the gain as a function of the sun position (solar azimuth and elevation angles) would

allow the reader getting a better insight about the operation and gain of the reconfigurable series-parallel PV module.

The following figure and analysis have been included:

[...] The DC yield difference between both modules is further evaluated in Figure 6 as a function of the solar position. Warm colours indicate when the yield of the reconfigurable PV module was higher than that of the reference PV module. From Figure 6a it is clear that in the absence of shading, the reference module performed better than the reconfigurable module during most of the experiment. Instead, during experiments when shading objects were mounted on the rack, the reconfigurable module outperformed the reference module except in two clear cases: (1) when the sun was in front of the module (i.e., around South in Experiment 1, and around Southeast in Experiments 2 and 3) because the shading objects on the sides of the modules were not causing shading, and (2) when the sun was behind the plane of array because the modules only received diffuse and reflected irradiance and the current mismatch was significantly reduced. [...]

Reviewer #2

The Authors wish to thank the Reviewer for their valuable comments and recommendations aimed at improving the quality of the manuscript.

A detailed reply to the comments of Reviewer #2 follows here below. Reviewer's comments are in black colour whereas Authors' replies are in red colour.

1. What is the LPVO MP1010F-1? I could not find in the reference [21] a full description of the hardware. Is it a dc-dc converter or what? More technical details are necessary.

The system used to monitor the reference PV module is a combination of a PV monitoring unit (PVMU) (i.e. an I-V tracer) and a DC/DC converter with MPPT (LPVO MP1010F-2, notice that there was a mistake in the device model that has been corrected in the paper). The technical datasheet of the equipment can be found in the brochure that can be downloaded from reference [21]. The authors have also included Table 4 in the Methods section with additional information about the monitoring equipment.

2. The reference module is connected to the LPVO and the reconfigurable module to a DC load. For sake of comparison is very important to use the same system and same measurements. The behavior of the MPPT, for example, can affect considerably the results.

The authors agree in ideal conditions the same equipment should be used to monitor both PV modules. However, this was not possible. First, the voltage and current range of the LPVO unit and any other off-the-shelf devices are not compatible with the voltage and current ranges of the reconfigurable module. Second, even if two identical DC loads would have been used for monitoring the reconfigurable and reference modules, both modules operate in different current and voltage ranges most of the time, hence measurements would have anyways with different levels of accuracy.

In addition, it is worth noting that as mentioned in the article, the power generated by both modules - which was also used to calculate the generated energy in the Results section - was not obtained from MPPT readings. Instead, it was extracted from the measured I-V curves. Therefore, the behaviour of the MPPT, which is active in between I-V curve measurements, only affects the module temperature. Although the module temperature does have a limited effect on the measured I-V curves, the authors corroborated that the power extracted by the MPPT algorithm of the LPVO tracking unit and the DC load were very close to the actual MPP extracted from the I-V curve. This implies that both PV modules were operating very close to the actual maximum power point and hence to the minimum possible temperatures.

Finally, more details about the measurement setup are now included in the manuscript to increase the confidence in the accuracy of the results:

[...] Both the REC and the REF PV modules were connected to the respective measurement equipment using 4-wire connections. Although it was not possible to monitor both modules using the same equipment due to the differences in the electrical output ranges, the selected equipment has similar measurement accuracy as shown in Table 4. The maximum measurement error was 7.5 mW and 5.5 mW with the BK8616 and the LPVO PVMU, respectively. [...]

3. Furthermore, in order to use a single module in a real application it is necessary a dc-dc stage in the inverter. This can affected the measurements and the behavior of the reconfigurable matrix. For a definitive validation of the proposal, it is important to use a system similar to the one that will be used in actual applications.

The presented reconfigurable module was designed to operate with a dedicated DC/DC conversion stage, which should be connected to the output of the reconfiguration matrix. This is similar to the way in which strings of PV modules operate with power optimizers, where the DC/DC stage in each power optimizer implements the MPPT capability and facilitates the connection of the PV modules to the inverter.

In the presented experiments, the output of the switching matrix was connected to a DC load, which was configured to maintain the reconfigurable PV module at its maximum power point (using an MPPT algorithm) in between regular I-V sweeps. The only difference between the experimental setup and how the module would operate in real conditions is that the DC/DC stage had to be substituted with a DC load to monitor the performance of the system. Therefore, the operating conditions of the switching matrix during the experiments are similar to a fully-fledged implementation of the reconfigurable module.

Furthermore, it should be noticed that the switching matrix is operated independently from and thus not affected by the DC/DC conversion stage. On the contrary, it is the operation of the switching matrix that would affect the operation of the DC/DC conversion stage. For this reason and as mentioned in the article, the reconfigurable PV module requires the development of a special DC/DC converter (which is out of the scope of this work). This special DC/DC converter should be able to perform with high efficiency in the discussed voltage and current ranges and should also be able to tolerate the brief power supply interruptions at each reconfiguration event.

4. It is said that a microcontroller is used for the reconfiguration algorithm, but according to Fig. 4 it seems that a PC is used.

A microcontroller was indeed used to execute the reconfiguration algorithm. The PC was only used to synchronize the reconfiguration events with the operation of the electronic load, which needs to receive a command to perform the I-V sweep. To avoid confusion, Figure 4 (now Figure 12) was modified and the following text was added:

[...] The measurement procedure for the reconfigurable module is shown in Figure 12. The setup consisted of a PC that was used to synchronise the operation of the switching matrix and the DC load, and to store the measurements. Reconfiguration was performed on a minutely basis by the microcontroller in the switching matrix upon the reception of the triggering command sent by the PC. [...]

5. Even if the energy consumed by the microcontroller, sensing circuitry, mosfets gates, etc. is low, it must be considered in order to a fair comparison.

The following discussion has been added:

[...] It is worth mentioning that the yield of the reconfigurable module in Table 1 is excluding the energy consumed by the switching matrix and the sensing circuitry, which is considered negligible. While the switching matrix can be kept in low-power consumption mode in between reconfiguration intervals reducing its consumption to 10 mW, the reconfiguration intervals last in average 150 ms during which the power consumption raises to about 1 W. Considering 16 hours of daylight and minutely reconfiguration events, it is estimated that the energy consumed by the electronics in the reconfigurable module was approximately 20 Wh for the entire monitoring campaign, which is about 0.02% of the total energy delivered by the reconfigurable PV module. [...]

6. The paper is as an experimental validation of [17]. [17] is much more complete and some important information could be put in the present paper

In addition to the corrections and additional information included throughout the revised version of the manuscript, the authors have also included the following:

- Figure 2 to facilitate the visualization and comprehension of the reconfigurable module.
- Figure 11 to explain in detail the logic behind the reconfiguration algorithm.
- The BoM of the PV modules and the electrical specifications of the used solar cells to improve the reproducibility of the results.

- All relevant measured data as well as code to facilitate its accessibility were uploaded in the public repository mentioned in the Data availability section.

REVIEWERS' COMMENTS

Reviewer #1 (Remarks to the Author):

Thanks for the revised manuscript and the response to my review. All issues raised in the first review have been addressed and the current version of the manuscript is from my point of view ready for publication.

Reviewer #2 (Remarks to the Author):

The paper has improved, but some points should be still considered:

- The function of LPVO MP1010F-1 is explained in the response letter, but I think it is not yet very clear in the paper.
- In Fig 6, the difference in kWh is related to which period? The period of measurement (4months)? Consider to use the percentage of gain/loss instead of direct energy.
- The values in Figure 7 and 8 are related to which period? One day, 4 months?
- It is important to describe the methods before the results
- The number of the Sections are missing and there is no Section of Conclusions.

Reviewer #2

The Authors wish to thank the Reviewer for their valuable comments and recommendations aimed at improving the quality of the manuscript.

A detailed reply to the comments of Reviewer #2 follows here below. Reviewer's comments are in black colour whereas Authors' replies are in red colour.

1. The function of LPVO MP1010F-1 is explained in the response letter, but I think it is not yet very clear in the paper.

The following explanation has been included in the Methods section:

[...] The electrical performance of the reference PV module (REF) was monitored using an LPVO PV monitoring unit (PVMU) in combination with an MP1010F-2 MPPT unit. The PVMU was used to perform I-V sweeps on the reference PV module at every minute. Meanwhile, the MP1010F-2 MPPT unit was employed in between these sweeps, ensuring a continuous extraction of power from the reference module by operating it at its maximum power point. [...]

2. In Fig 6, the difference in kWh is related to which period? The period of measurement (4months)? Consider to use the percentage of gain/loss instead of direct energy.

Each subfigure in Figure 6 (now Supplementary Figure 1) was generated for the full length of the corresponding experiment detailed in Table 1. The authors have now included this clarification in the caption of the figure.

In addition, the authors have evaluated showing the gain/loss results in relative terms, as suggested by the reviewer. However, it was considered that if the values were presented in percentages rather than kWh, the high relative percentage gains (~50%) obtained for some specific azimuth-altitude bins could confuse the reader, given the lower relative DC yield difference values in Table 1, which apply to each experiment as a whole.

3. The values in Figure 7 and 8 are related to which period? One day, 4 months?

Again, the results correspond to the full periods detailed in Table 1. The authors have now included this clarification in the figure captions.

4. It is important to describe the methods before the results and 5. The number of the Sections are missing and there is no Section of Conclusions.

The name, format and order of the sections have been generated according to Nature Communication's editorial guidelines and must remain as such as instructed by the editor.